# Health behaviors and care seeking practices for childhood diarrhea and pneumonia in a rural district of Pakistan: A qualitative study

Jai K. Das[1,2]*, Faareha Siddiqui[3], Zahra Ali Padhani[4], Maryam Hameed Khan[1], Sultana Jabeen[2], Mushtaq Mirani[1], Shaista Mughal[2], Shafaq Baloch[2], Imtiaz Sheikh[1], Sana Khatoon[1], Khan Muhammad[1], Manesh Gangwani[2], Karim Nathani[2], Rehana A. Salam[5], Zulfiqar A. Bhutta[1,6]

1 Institute for Global Health and Development, Aga Khan University, Karachi, Sindh, Pakistan, 2 Division of Women and Child Health, Aga Khan University, Karachi, Sindh, Pakistan, 3 Bloomberg School of Public Health, Johns Hopkins University, Baltimore, MD, United States of America, 4 University of Adelaide, Adelaide, SA, Australia, 5 Melanoma Institute Australia, Centre of Research Excellence, University of Sydney, Camperdown, Australia, 6 Centre for Global Child Health, The Hospital for Sick Children, Toronto, ON, Canada

* jai.das@aku.edu

**Data Availability Statement:** All data generated or analyzed during this study are included in the paper.

## Abstract

Diarrhea and pneumonia are the leading causes of morbidity and mortality in children under five, and Pakistan is amongst the countries with the highest burden and low rates of related treatment coverage. We conducted a qualitative study as part of the formative phase to inform the design of the Community Mobilization and Community Incentivization (CoMIC) cluster randomized control trial (NCT03594279) in a rural district of Pakistan. We conducted in-dept interviews and focused group discussions with key stakeholders using a semi-structured study guide. Data underwent rigorous thematic analysis and major themes identified included socio-cultural dynamics, community mobilization and incentives, behavioral patterns and care seeking practices for childhood diarrhea and pneumonia, infant and young child feeding practices (IYCF), immunization, water sanitation and hygiene (WASH) and access to healthcare. This study highlights shortcomings in knowledge, health practices and health systems. There was to a certain extent awareness of the importance of hygiene, immunization, nutrition, and care-seeking, but the practices were poor due to various reasons. Poverty and lifestyle were considered prime factors for poor health behaviors, while health system inefficiencies added to these as rural facilities lack equipment and supplies, resources, and funding. The community identified that intensive inclusive community engagement and demand creation strategies tied to conditioned short term tangible incentives could help foster behavior change.

## Introduction

In 2020, an estimated five million children under the age of five died [1]. Despite a 61% decrease in under-five mortality in the last three decades, majority is still attributable to

**Funding:** The study was funded by Bill and Melinda Gates Foundation under grant number OPP 1148892. The funders had no role in study design, data collection and analysis, decision to publish, or preparation of the manuscript.

**Competing interests:** The authors declare that the research was conducted in the absence of any commercial or financial relationships that could be construed as a potential conflict of interest. This does not alter our adherence to PLOS ONE policies on sharing data and materials.

communicable and infectious diseases that are both preventable and treatable. Globally, pneumonia, diarrhea and malaria continue to remain the leading causes of death [1] and this burden is disproportionately borne by the poorest regions. Nigeria, India, and Pakistan were the top three countries with the highest death toll in 2019 [2].

Diarrhea and pneumonia are regarded as diseases of poverty, associated with undernutrition, environment, and lack of access to healthcare [3]. Mortality is preventable by appropriate nutrition including infant and young child feeding practices, vaccination, proper hygiene and sanitation, and appropriate care. However, despite being readily preventable, pneumonia was responsible for 800,000 deaths, and diarrhea claimed the lives of 437,000 young children worldwide in 2018 [1]. Approximately 80% of the deaths in low- and middle- income countries (LMICs) occur due to diarrhea and pneumonia, of which 50% take place in India, Nigeria, Pakistan, and Ethiopia alone [4].

In 2015, United Nations adopted the Sustainable Development Goals (SDGs) which aimed to promote healthy lives and well-being for all children, with SDG-3 specially focused on ending preventable deaths of under-5 children by 2030 [5]. In 2013, the Integrated Global Action Plan for Pneumonia and Diarrhea (GAPPD) called for an integrated approach to protect, prevent, and treat diarrhea and pneumonia with interventions such as exclusive breastfeeding (EBF), timely complementary feeding, vitamin supplementations, vaccine coverage and water, hygiene, and sanitation [6].

Pakistan is among the countries with the highest under-five mortality [7,8] with pneumonia and diarrhea the leading cause [9]. Given the current statistics, Pakistan may not be able to achieve the SDG 3 target by 2030 [10]. According to the Pakistan Demographic and Health Survey 2017–8 [11], the prevalence of diarrhea among children under five has decreased from 23% in 2013 to 19% in 2017–18. However, lower coverage for preventive and therapeutic interventions persists, with care-seeking at 71%, early initiation of breastfeeding at 20%, EBF for 6 months at 48%, timely complementary feeding at 54% [11], and ORS given to only 37% of children with diarrhea. The prevalence of acute respiratory infection (ARI) was 14% in 2018 with about 84% of children taken to a health facility and 46% treated with antibiotics. While vaccine coverage for children aged 12 to 23 months has improved from 1991 (35%) to 2018 (66%) [11], it is still well below the 90% target set by GAPPD [11,12]. Despite existing interventions, uptake remains low as this requires change in behaviors and care seeking practices.

Health-related behaviors are often resistant to change and are influenced by a variety of personal, cognitive, economic, socio-cultural, and structural factors [13]. Over the past few decades, there has been a greater understanding of health-related behaviors both at individual and community level. The use of formative research has also increased to aid the implementation of culturally and geographically relevant intervention programs [14]. Various forms of incentives (including user fees reduction and conditional cash transfer) have been evaluated for their effectiveness and have shown the potential to improve coverage of evidence-based interventions [15–21].

There is a need for action to reduce preventable child morbidity and mortality. We conducted formative research to inform the design of a cluster randomized controlled trial, by exploring possible incentives which could improve community engagement and active participation to influence behavior change in a rural district of Pakistan. Through qualitative assessment, we attempt to understand the local community's knowledge and beliefs, water, sanitation, and hygiene conditions, IYCF and immunization practices, identify current care seeking behaviors for childhood diarrhea and pneumonia and highlight the possible barriers and facilitators in the uptake of essential interventions.

## Materials and methods

### Study design

This qualitative study was conducted as part of the formative phase to inform the design of the Community Mobilization and Community Incentivization (CoMIC) cluster randomized control trial (NCT03594279) in the Tando Muhammad Khan (TMK) district of Sindh, Pakistan [22]. The CoMIC trial is a prospective three arm cluster randomized trial with two intervention groups and a control group, which aimed to assess the effect of a community engagement and demand creation strategy with a conditional collective community-based incentive (C3I) to improve care seeking practices and adherence to prevention and management practices for diarrhea and pneumonia in children under five years of age.

We conducted a series of qualitative investigations at baseline to identify and assess the feasibility of communication channels and potential non-cash incentives that would be beneficial to the community. The in-depth interviews (IDI) and focus group discussions (FGDs) also aimed to understand the socio-cultural dynamics, behavioral patterns, care seeking practices and context-specific barriers and facilitators, and adherence to recommended practices for the prevention of diarrhea and pneumonia. Findings from this study were built in the communication strategies of the CoMIC trial.

### Study setting

Sindh is the second largest province by population (approximately six million people) and the third largest province in Pakistan. The study was conducted in the district of TMK, one of the 29 districts in Sindh, which has an area of 1,814 km$^2$ and a population of approximately 677, 228. Children aged 0–14 years account for 43% of the population [23]. The district has a poverty rate of 78.4% [24]. It comprises of three administrative talukas and 17 union councils (UCs). The TMK district has a population density of approximately 75.8/km$^2$ and an annual population change rate of approximately 2.3%, it is also a representative of any rural district in Sindh.

### Participants and recruitment

The CoMIC research team was already present on site before the study commenced, carrying out informal discussions for the purpose of the study within the community and identifying potential stakeholder groups. Study participants were recruited through purposive sampling by social mobilizers utilizing community networks and ensuring ethnic and geographical diversity and representation from all areas of TMK. Snowballing technique was used to identify further participants from the various identified groups, including healthcare professionals (doctors from the public and private sector, lady health workers (LHWs), nurses and midwives), community representatives, policy makers, religious leaders, government officials (district administrators) and parents of children under five. Participants were mostly approached face-to-face except for some government officials who were contacted via telephone, while all IDIs and FGDs were conducted face-to-face in the community or offices of the health care providers and government officials. To maintain gender balance, both male and female participants were interviewed in each identified stakeholder group.

### Study guide

The research team conducted a relevant literature review on community engagement, diarrhea, and pneumonia followed by discussions within the team and the community, and incorporation of their input to finalize probing questions. Subsequently, a semi-structured interview guide was developed in English with predetermined open-ended questions. We also

**Table 1. Total number of interviews.**

| FGDs | Total FGDs conducted | IDIs | Total IDIs conducted |
|---|---|---|---|
| Father | 4 | Father | 3 |
| Mother | 3 | Mother | 3 |
| Community leaders and UC members | 3 | Community elder | 3 |
| LHW | 3 | LHW | 2 |
| Nurse | 1 | Nurse | 1 |
| | | LHS | 2 |
| | | CHW | 1 |
| | | Government physician | 3 |
| | | Private physician | 3 |
| | | Policy maker | 2 |
| **Total** | **14** | **Total** | **23** |

*Acronyms*: Union council (UC); lady health worker (LHW); lady health supervisor (LHS); community health worker (CHW); in-depth interviews (ID); focus group discussion (FGDs).

received feedback from researchers with long-term experience working in the community, to ensure applicability and relevance of the interview questions. The guide was then translated into the local Sindhi language and pilot tested on five individuals. The guide included questions pertaining to community characteristics, childhood diseases particularly diarrhea and pneumonia, childhood vaccinations, infant and young child feeding (IYCF), water, sanitation, and hygiene (WASH) conditions, healthcare seeking behaviors, health systems, possible mediums of community mobilization and views regarding conditional incentives.

## Data collection

A total of 23 IDIs and 14 FGDs (with 134 individuals) (Table 1) were conducted by trained researchers with experience working in community and conducting qualitative research. A defined number of participants were included, there were no refusals or dropouts, and no repeat interviews were conducted. The total duration of each qualitative interview was approximately 40–60 minutes and audio recordings were made using a digital voice recorder. Interviews were conducted in the local language Sindhi. IDIs and FGDs were conducted in the community or offices of healthcare providers and government officials and were held in as much privacy as possible. No other person apart from the participants and research team were present during the consultations. The interviewers also made extensive notes during each IDI and FGD. At the conclusion of each day of data collection, the research team compiled their IDI and FGD field notes, all audio recorded IDIs and FGDs were transcribed verbatim in Sindhi, and then translated into English. Sections of narrative were translated and back translated for quality assurance. English translations were rechecked to identify discrepancies.

## Data analysis

Following data collection, thematic analysis was conducted using NVivo software (version 10) to manage and code interviews [25]. We used the thematic approach developed by Braun and Clarke [26] which included the seven steps of transcription/translation, reading and familiarization, coding, searching for themes, reviewing themes, defining, and naming themes and finalizing the analysis. FS, MHK, SJ and ZAP read and reread through each transcript to gain familiarity with the content and context. Inductive coding was performed by each researcher on the first four interviews to develop a coding scheme which was subsequently reviewed,

discussed, and finalized, followed by coding of the remaining transcripts. Responses were coded to relevant nodes, which were later categorized into a hierarchy of tree-nodes. Once coding of the entire dataset was completed, emerging themes and subthemes were drawn from the tree nodes which would help to answer our research objectives. Two levels of themes emerged from the data; 'childhood diseases', 'access to health facilities and providers', and 'health systems' were further categorized into the relevant subthemes. The process was completed with each translation being reviewed to ensure that all necessary information had been captured and appropriately listed, categorized, and interpreted. Relevant quotations were extracted from the interviews and included to relay critical findings.

### Rigor

A field office was set up prior to commencement of the study, with the research team engaging with the community and carrying out informal discussions over an extended time to foster collaboration and trust. The IDIs and FGDs were conducted by JKD, MM, SB, IS, KM (male) and SJ, SM (female). All researchers were formally trained, with extensive experience in field work and conducting IDIs and FGDs within communities. Triangulation of sources was achieved by interviewing both men and women participants across diverse groups in TMK, ranging from local community members, healthcare professionals, religious leaders, and government officials. During the data analysis process, each transcript was read, re-read, annotated where relevant information was found, and analyzed separately by four researchers. Verifying the accuracy of emergent codes, themes, and interpretations was achieved by constant rereading, comparing, and redefining of themes both alone and between the four researchers to ensure intra and inter-rater reliability. Discussion of the researchers' interpretations were carried out with two researchers not directly involved with the study. Low-inference descriptions i.e., direct quotations from study participants were included to relay critical findings.

### Ethics

Ethical approval was obtained from the Aga Khan University's Ethical Review Committee (AKU-ERC) (ERC#: 4676-WCH-ERC-17). The study's consent form was read and explained in detail, and written informed consent was taken from all participants prior to the interviews. Through the consent form, participants were introduced to the researchers, explained the background, purpose and procedure for conducting the study. The respondents were informed about their rights. Voluntary participation was ensured, and participants had the autonomy to refuse questions or leave the interview at any point. They had the right to ask questions at any point before, during or after the interview. All interviews were conducted by trained personnel and in conditions of privacy. Confidentiality was maintained by using auto-generated participant codes instead of names.

## Results

A total of 157 participants were included in the 23 IDIs and 14 FGDs. Table 2 summarizes sociodemographic characteristics of the participants. Major themes emerging from the IDIs and FGDs are detailed in Fig 1.

### Community characteristics and social networks

The population of TMK is comprised of Muslims and Hindus of different ethnicities, tribes, and castes. Sindhi is the predominant ethnicity, while minorities include Punjabi, Balochi, Pashtuns, and Muhajirs. The tribe defines the basic identity of individuals and groups, and each tribe has local elders or leaders who are consulted on important issues and are responsible

**Table 2. Characteristics of study participants.**

| Study participant characteristics (N: 157) | |
| --- | --- |
| | **N (%)** |
| **Age** | |
| 20 to 40 | 121 (76) |
| 41 to 60 | 36 (23) |
| **Education** | |
| Primary | 10 (6) |
| Matriculation | 12 (8) |
| Intermediate | 29 (18) |
| Bachelors | 35 (22) |
| Masters | 3 (2) |
| Religious education | 1 (1) |
| No education | 66 (41) |
| **Occupation** | |
| Unskilled labor | 38 (24) |
| Salesperson | 6 (4) |
| Landowner | 3 (2) |
| Academia | 1 (1) |
| Health professional | 46 (29) |
| Religious scholar | 4 (3) |
| Community representative | 20 (13) |
| Home maker | 26 (16) |
| **Income (PKR)** | |
| 5000 to 15,000 | 55 (35) |
| 16,000 to 25,000 | 46 (29) |
| 26,000 to 35,000 | 2 (1) |

*Acronyms*: Pakistani Rupee (PKR).

for making major decisions for the community. Participants explained that there are strong tribal associations among members belonging to the same tribe. Several languages are spoken, with Sindhi being the most common. The population, although diverse, has religious and ethnic tolerance and lives in harmony.

> *"In this locality both Muslims and Hindus reside and belong to different tribes. People speak different languages such as Sindhi, Siraiki, Balochi, Pashtu, Dhakti and Hindi"-* *(IDI-TMK-M1)*

Most families belong to a low socioeconomic background with limited access to basic amenities. Housing and living conditions are poor, with people usually living in large joint families and residing in constricted spaces. Education and employment opportunities are limited, and most residents work as unskilled laborers in farming and construction. While efforts to improve education have been made by non-governmental organizations (NGOs), the overall literacy rate is low in the community.

## Community mobilization

Currently, community mobilization efforts are variable across different villages in TMK, and in some villages are entirely nonexistent. The community identified an imminent need for

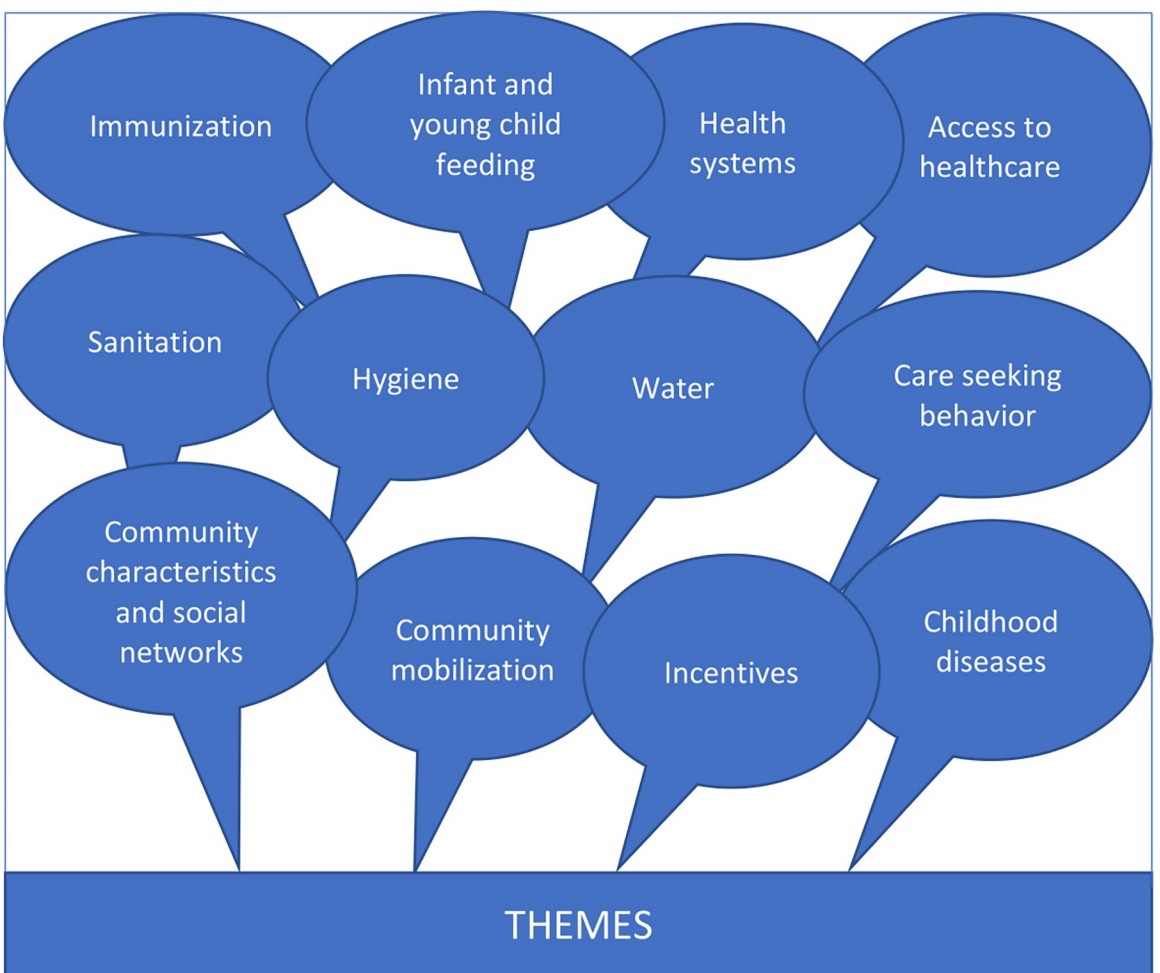

**Fig 1. Major themes from IDIs and FGDs.**

improvement in health and hygiene behaviors and practices and believed mobilization programs can create awareness by educating local people. The most important issues in the community were identified to be the prevention and treatment of diarrhea and pneumonia, water and sanitation, improvement in breastfeeding practices and immunization. There was a unanimous opinion that forming community support groups would be the most efficient and socially acceptable way to convey messages to the greatest number of people.

*"In my knowledge, there are no community groups or organizations working to improve situation in Bulri Shah Karim"- (IDI-RHS-N1)*

Participants showed significant interest in developing community groups which can be used to promote healthy practices and solve communal problems. They suggested that these groups may be in the form of village health committees, unions, or local organizations (*tanzeem*). Keeping in view the cultural and social characteristics of the community in TMK, participants believed that tribal differences and the lack of unity between community members can potentially hamper efforts to mobilize and create change. Most importantly, each group should have members with similar tribal identity and affiliations. To further promote the

acceptability of group decisions and encourage positive outcomes, it was suggested that groups should be carefully formed to include able people who have influence and respect in the community such as religious and tribal leaders, feudal lords, community elders and representatives, health providers and teachers. This would lead to more effective problem solving while enhancing organizational capacity and ensuring representation of all voices within the community.

*"We formed a community group and have included influential people such as an imam (religious leader), landlords, shopkeepers, and other respected members of the community. This helps us convince people that work we are doing is for their betterment"- (IDI-RHS-LHS1)*

The community of TMK eagerly suggested that mobilization and participatory learning is the way to change, and UC members should be involved in long term awareness programs. In many villages, communal gathering spaces are limited, and participants recognized the foremost need for community centers to facilitate group meetings. Children were identified as the 'agent of change' especially when it came to sanitation and hygiene. Innovative methods of message communication were suggested, especially visual communication including video and multimedia (brochures, pictorial messages, and flip charts). Participants recalled the success of pictorial charts for breast feeding practices which aided quick learning because women in the community found them interesting and relatable. These methods would be most effective, especially keeping in mind the low literacy rate and inability to read.

## Incentives

We explored various incentivization schemes with the community of TMK. Although general community preference leans toward unconditional incentives, when we discussed the potential of conditional non-cash incentives, participants agreed that this would be the best way to successfully promote healthy behaviors and practices in TMK. The community believed that if they were in charge of deciding what incentives they would receive after fulfilling the required conditions, there would be greater acceptance and overall success of the incentivization scheme.

Non-cash incentives that were most popular among the participants and could have the potential to encourage positive behavioral change included construction of toilets and latrines, installation of hand pumps to access water, and distribution of hygiene kits to facilitate hand washing and bathing. Some community members also demanded the construction of hospitals, animal sheds, roads and sewage lines, and provision of food. They stated that because these non-cash incentives would be relevant to the problems currently faced by the community and could lead to a significant improvement in quality of life, they would appeal most to the local people and will likely be a prominent driver of behavior change.

*"Installation of hand pumps will attract the community to improve personal hygiene and hand washing"- (FGD-EM-M1)*

When asked whether the community would contribute to these incentives, most declined monetary contribution, but agreed to invest time, land, and labor. This would, in effect, stimulate the residents to become involved in community activities, foster a sense of ownership and promote solidarity. As opposed to incentives at an individual level, participants were instead keen on an incentivization initiative that would motivate the community to work in unison towards serial targets and rewards, as this would be most successful in improving overall health-related behaviors and practices.

*"If we install water filter plants, and developments are observed by the people in villages, they can be motivated to adopt hand washing"–(FGD-TGH-4)*

## Childhood diseases

**Diarrhea.**   Although the community was aware of the symptoms, causes and preventive measures such as maintaining personal hygiene and a clean environment, diarrhea remains one of the most prevalent issues in TMK. To combat this problem, participants suggested a need for improvement in health behaviors and practices. This included increasing awareness of hand washing, consumption of safe drinking water, preventing children from playing in unclean areas or walking barefoot and encouraging EBF and timely vaccination. At present, the local people seek care from healthcare providers including doctors and LHWs who prescribe antibiotics and IV fluids. However, numerous misconceptions are still at play in the community. Some mothers stop breastfeeding as either they consider it harmful for the child during episodes of diarrhea, or they are pressurized by traditional norms and beliefs. There is also a lack of awareness regarding the benefits of ORS, resulting in low usage of ORS during diarrhea episodes. Additionally, traditional healing methods which may not be beneficial are regularly sought by the community. Details of knowledge, practices and interventions are presented in Table 3.

*"Mostly they use traditional methods, like mint water, sago, banana and yogurt, to treat children suffering from diarrhea."*

Healthcare providers and district administration officials discussed that the overall prevalence of diarrhea has reduced compared to previous years, which can be attributed to timely

**Table 3.  Knowledge, practices and interventions for diarrhea and pneumonia.**

| | Knowledge | Practices | Interventions |
|---|---|---|---|
| **Diarrhea** | ▪ Most participants were aware of:<br>  ◇ Causes and symptoms<br>  ◇ Prevention by clean environment and good personal hygiene<br>▪ Treatment of diarrhea through ORS and oral or intravenous antibiotics<br>▪ Misinformed belief that breastfeeding should be withheld during diarrhea as it may be harmful for the child. | ▪ Customary for certain foods to be given or withheld during diarrheal episodes. People give soft diet such as *khichri* (lentils and rice), *sabudana* (tapioca pearls), yogurt, and bananas. Spicy and oily foods and snacks are avoided.<br>▪ Traditional and spiritual healing methods are also used. People visit religious scholars and use remedies such as drinking cardamom and mint water and applying onion juice to the armpits and face.<br>▪ The use of ORS is low due to lack of awareness, knowledge, and trust. | ▪ ORS, Zinc supplements and Metronidazole are commonly prescribed treatments.<br>▪ Intravenous medication is given as per requirement.<br>▪ ORT corners constructed to treat mild to moderate dehydration under supervision of trained staff.<br>▪ Timely vaccinations and inclusion of Rota vaccine in the EPI.<br>▪ LHWs efforts in conducting awareness sessions, and involvement in distributing ORS and zinc supplements to households. |
| **Pneumonia** | ▪ Participants used the terms pneumonia and asthma interchangeably. They were aware of:<br>  ◇ Symptoms including cough, fever, chest congestion and pain, breathing difficulties, and lungs filled with pus and fluid.<br>  ◇ Prevention measures that ensure children remain warm during winter.<br>▪ Pneumonia is considered a seasonal disease that increases during winter.<br>▪ People are less equipped to deal with pneumonia as compared to diarrhea. | ▪ Mothers often take their children to work in open fields without covering them in appropriate warm clothing during winter.<br>▪ Majority rely on treatment from doctors, as LHWs are not qualified to treat pneumonia and make referrals to doctors.<br>▪ Traditional methods of healing such as giving honey, putting balms on throat, ribs, and chest | ▪ Timely vaccination<br>▪ Antibiotics and nebulization were given depending on severity of disease.<br>▪ LHWs conducting awareness campaigns regarding better health practices. |

*Acronyms*: ORS- Oral Rehydration Salts; ORT- Oral Rehydration Therapy; EPI- Expanded Program on Immunization; LHW: Lady Health Worker.

vaccinations (especially the rotavirus vaccine), presence of doctors at health facilities, awareness sessions and distribution of ORS and zinc supplements by LHWs. However, further efforts are required to improve the current practices of the community in TMK.

**Pneumonia.** Although participants were aware of the symptoms, and prevention techniques, many community members used the terms pneumonia and asthma interchangeably, signifying a significant gap in knowledge. Due to financial conditions and the predominant labor employment in TMK, most parents accepted noncompliance with routine preventive measures for their children. They identified that timely vaccination (mainly the pneumococcal vaccine) should be encouraged. Details are presented in Table 3.

Majority of the people in the community rely on treatment prescribed by doctors. Children are usually nebulized and given antibiotics depending on the severity of disease. However, allopathic treatment is supplemented by traditional remedies such as honey and applying balms on the throat and chest.

Efforts have been made to reduce pneumonia related morbidity and mortality by providing free vaccination, increasing referrals and providing appropriate treatment at government hospitals. However, due to the lifestyle and employment conditions in rural TMK, pneumonia remains a prevalent issue for the community requiring further socially and culturally relevant interventions.

There appeared to be similar levels of knowledge, practices and health seeking behaviors for both diarrhea and pneumonia in TMK, allowing the implementation of a common intervention to target both illnesses and improve outcomes in the community.

## Infant and Young Child Feeding Practices (IYCF)

Fig 2 depicts issues related to breastfeeding in TMK. Due to cultural practices in the community, participants discussed the custom of bathing infants after birth, resulting in a significant delay in breastfeeding initiation. Most mothers were unaware of the concept and benefits of EBF and routinely give water and other traditional foods to their infants, resulting in minimal EBF.

Over time, efforts made by doctors and LHWs to increase awareness and dispel misconceptions has led to a decrease in the prevalent tradition of discarding colostrum. Now, some mothers breastfeed their infants when they have diarrhea and pneumonia and are aware that breastfeeding should be continued for two years. Complementary feeding should ideally be started after six months, but most people in TMK do so after the four months of birth.

*"Exclusive breastfeeding is not practiced in our culture. We give honey, sweet paste, and water during the first six months of life"- (IDI-TMK-F1)*

There is a clear need for efforts to improve current IYCF practices. The community believed that awareness programs and village committees would be useful in educating women about EBF, colostrum feeding and family planning. They suggested that the elders of the tribes should also be made aware of the benefits of colostrum, and to promote acceptability and uptake of improved health behavior, doctors, LHWs, religious leaders, and landlords should be involved in educating the masses. Additionally, health programs designed to improve the health and nutrition of women in rural TMK should be initiated.

## Immunization

Community members receive vaccinations by visiting hospitals, rural health centers, vaccination centers, or by home visits from vaccinators and LHWs. Vaccines and government stamped vaccination cards are provided free of cost.

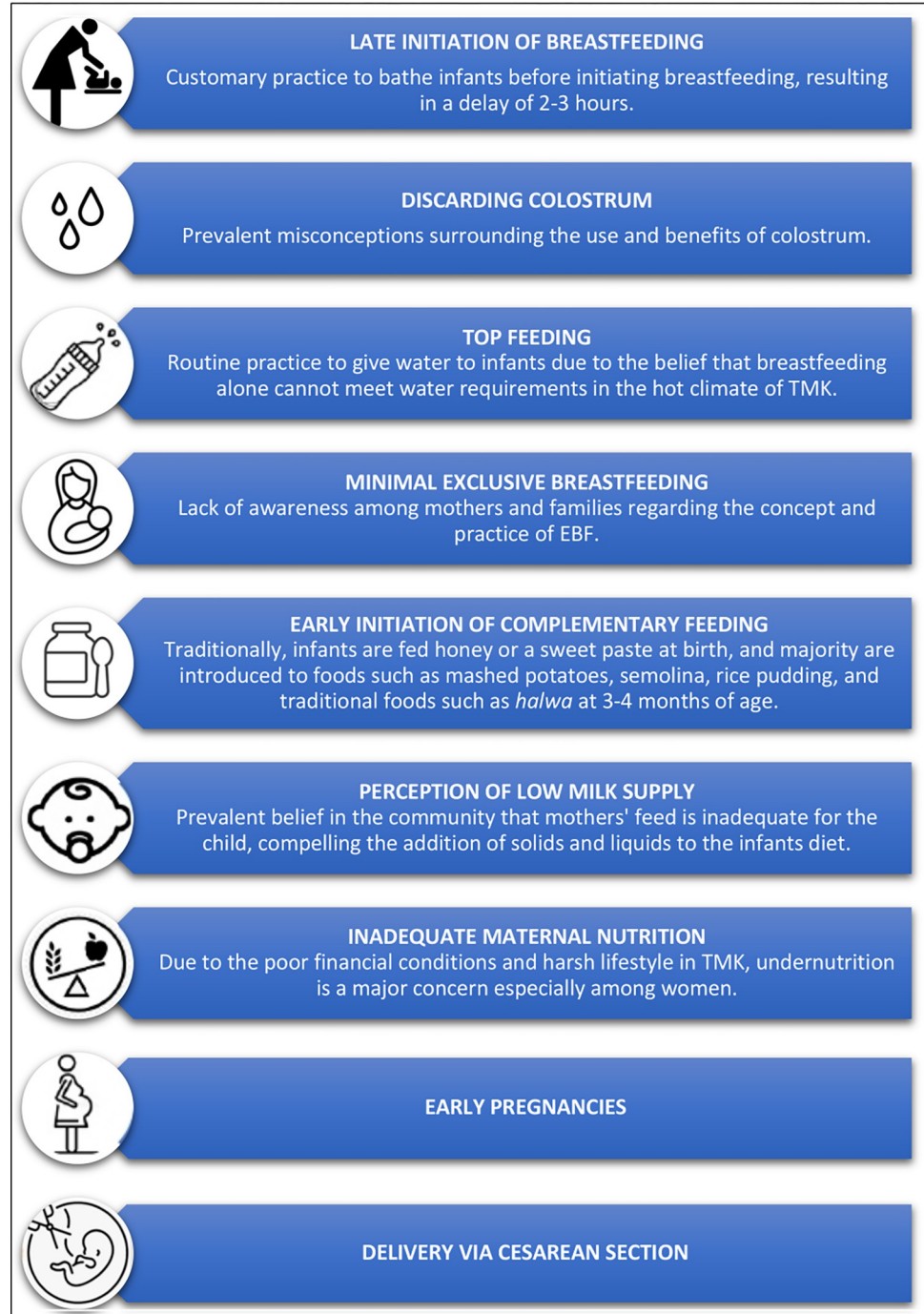

**Fig 2. Issues related to breastfeeding.**

The local people were aware that vaccines are mandated and can reduce the prevalence of diarrhea, pneumonia, and other diseases. Although most children have been vaccinated, issues in complete and timely vaccination persist, largely due to logistical and behavioral reasons. Only a few participants were aware that rotavirus vaccine was added to the vaccination schedule and can prevent diarrhea, underscoring the need for awareness efforts in the community.

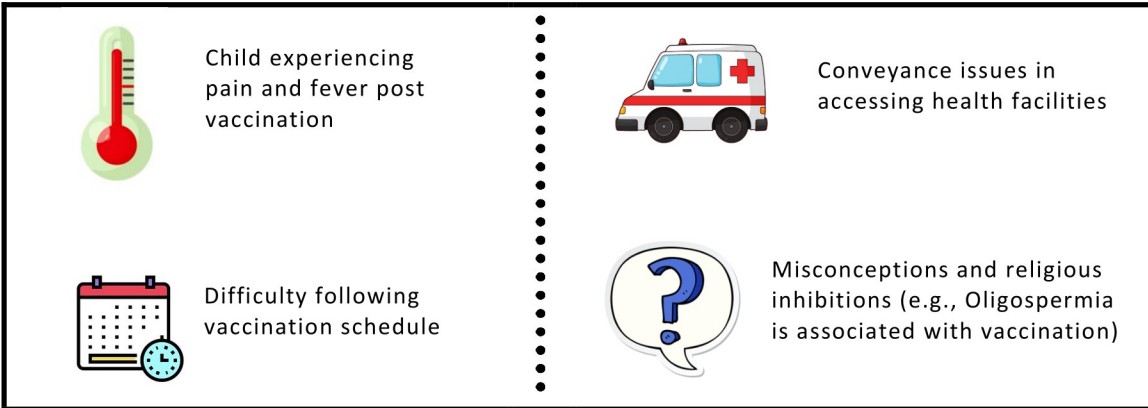

**Fig 3. Issues related to immunization.**

*"After the death of some children as a result of diarrhea, mothers are making an effort to get their children vaccinated"—FGD-BSK-LHW6*

Most residents rely on home visits by LHWs and vaccinators, as every family does not have spousal and familial support in travelling long distances to access vaccination centers. Unfortunately, the visits are irregular, and vaccinators do not visit every village. There are challenges on the supply side as well, with closure of health facilities and absence of staff on the day specific vaccines must be received.

*"No, vaccinators are not enough. In our area of Shah Kareem, we only have two vaccinators"-(IDI-RHS-N1)*

Fig 3 depicts reasons for vaccine refusal. While efforts have been made to discredit misinformation and myths, more still is required to reduce vaccine hesitancy in the community. Suggestions from the local community included participants being informed about the importance of vaccination and its role in preventing diseases. Community mobilization should be encouraged and committees including religious scholars should be formed to dispel myths. Vaccinators should inform the local population about potential side effects, reiterate that symptoms such as pain and fever are not a cause for concern, and provide pain medication if required.

## Water, Sanitation and Hygiene (WASH)

Water is primarily obtained using unsanitary hand pumps (individual and communal) and bore holes, making unclean water a major problem for the community. The limited water supply affects personal hygiene, preventing regular handwashing and bathing. This in turn increases susceptibility to infectious diseases. Participants identified gastrointestinal problems such as diarrhea, cramping and flatulence, skin, and eye diseases as common problems resulting from unclean water. Testing by authorized agencies deemed the water, containing a high quantity of total dissolved solids (TDS), unfit for human consumption. Most rural households do not have proper water storage mechanisms, and many residents were unaware of water treatment methods or the consequences of using untreated water. Others who were, were not practicing it due to the lack of time and resources. We discussed methods of improving water

quality and supply with the local community, who suggested increasing awareness of the benefits of water treatment, provision of water tanks for proper storage, and installation of filter plants, particularly the Reverse Osmosis (RO) plant.

Sanitation is dismal in the rural villages of TMK. We were told that most people do not have access to toilet and latrine facilities and practice open defecation, which pollutes the environment and increases the risk of health problems. Although participants were aware of the resultant problems, including illnesses such as diarrhea, vomiting, hepatitis, polio, and skin problems, these practices persist. Inefficient waste disposal was another issue faced by the rural community. The municipal committee installs large dustbins at selected disposal points and sweepers are responsible for collection using municipal vehicles. Unfortunately, garbage collection is compromised since there are usually no large dustbins or formal waste collection points in villages. Residents complained that irregular service causes disposal points to be filled beyond capacity, and they are left to manage waste by dumping and burning it in canals and sewage lines. Villages have poor drainage systems. Where pipelines and gutters are installed, lack of maintenance results in frequent overflow. Dirty water is often stagnant on streets and is contaminated with animal and human feces. Being aware of these issues, the local community desired a significant improvement in sanitation and suggested the construction of toilets and separate animal sheds, development of more efficient drainage and garbage disposal systems, and sessions to increase awareness of hygiene practices.

Personal hygiene in TMK is inadequate due to irregular bathing and hand washing. The main reasons include unavailability of water, lack of awareness, extreme poverty, and rigorous work burden. According to the local people, initiatives such as awareness programs and distribution of soaps should be used to encourage regular hand washing. To improve current practices, community members should be informed about the importance of washing hands before cooking, ensuring that food is not undercooked, and keeping animals away from the kitchen. This will help maintain food hygiene and limit the spread of disease. Table 4 summarizes important findings related to WASH.

## Access to health facilities and providers

**Access to health facilities.**   Health facilities in TMK include Rural and Primary Health Centers (RHCs and PHCs), Basic Health Units (BHUs), government dispensaries and hospitals, and private hospitals/clinics. The community prefers to receive treatment from hospitals rather than traditional or spiritual healers. Majority of the local population seeks care at public hospitals because they are free of cost and are easily accessible. Since most residents of TMK live in poverty, free consultations and medicines were conveyed as the primary reason for visiting public hospitals.

*"Here, majority of the people are daily wagers, laborers in fields, brick furnaces, or sugar mills. Their income does not even cover their expenses so when they are faced with disease, they come to government hospitals. Here treatment and medicines are free, and there are no doctors' fees"—FGD-LHS1*

Many community members stressed that were they financially capable, they would opt to visit private facilities for multiple reasons. Public hospitals are piled with complaints such as absence of staff, unfriendly attitude of physicians, long waiting periods, lack of medical equipment and substandard quality of care. Additionally, government facilities are not functional after 2pm, and so are unable to tend to most medical emergencies in the area.

**Table 4. Findings related to WASH.**

|  | Issues | Quotations |
|---|---|---|
| **Water** | ■ Water scarcity.<br>■ No municipal water distribution network.<br>■ Severely compromised water quality (brackish, acidic, and contaminated with mud, feces, chemicals, and industrial waste).<br>■ Irregular water treatment practices in households (e.g., boiling, aqua tablets, phosphorous, alum, straining through a cloth, and leaving the water in mud pots for sedimentation). | *"We are faced with the major issue of water scarcity and unsafe drinking water"- (FGD-EM-M1)*<br>*"Our water has high levels of arsenic, five times more than the standard level. TDS and conductivity are also high. Therefore, the water is not drinkable. This is a major issue here. If measures are not taken to resolve this, then there will be significant health consequences"- (IDI-BSK-PD1)*<br>*"They say that they do not have sufficient time to boil the water, wait for it to cool, and then pour it into containers. They cannot do this because they are already consumed with work since they must work on the fields and take care of their household, children, and animals"- (IDI-TGH-LHW1)* |
| **Sanitation** | ■ Lack of toilet or latrine facilities.<br>■ Open defecation is commonly practiced.<br>■ People openly defecate because of water scarcity; open defecation requires less use of water than toilets.<br>■ Some people refuse to use community toilets and latrines because these facilities are often dirty due to unavailability of water.<br>■ Inefficient waste disposal.<br>■ Poor drainage system: roads and streets are flooded with dirty water. | *"People openly defecate in nearby fields, grounds, bushes, alleyways and in the backyard of their houses"—(FGH-LHS-1)*<br>*"No, there is no management for solid waste disposal. Government sweepers only show their faces on special days or during official visits"- (IDI-BSK-F1)*<br>*"When it rains approximately 1 to 2 feet of water accumulates on the roads outside houses. The water is mixed with animal waste. Children play in the dirty water"- (IDI-BSK-F1)* |
| **Hygiene** | ■ Poor personal hygiene.<br>■ Unsatisfactory hand washing practices due to lack of awareness and noncompliance.<br>■ Soaps are not used due to poverty and habit.<br>■ People choose not to wash their hands since they have very limited water supply.<br>■ Food hygiene requires improvement.<br>■ Domestic hygiene is compromised by people keeping herds inside their living spaces instead of in separately constructed sheds. | *"People living in rural areas are unaware of the importance of maintaining personal hygiene. They avoid bathing, brushing teeth, combing hair and children walk and play barefooted in muddy surroundings"- (IDI-RHS-LHS1)*<br>*"People do not pay attention to hand washing. They think that because their elders did not wash their hands and still lived a good life then why is there a need for these new practices to be followed?"- (FGD-LHV-2)*<br>*"There is no water. We must travel long distances to collect water. Imagine only having half a jug of water, do you think it is possible to use it for hand washing?"- (IDI-TMK PM1)*<br>*"No one has separate rooms here. People live together. Those who are informed keep their animals away, and those who are not keep them inside their homes."- (FGD-BSK-CL6)* |

*"We go to private clinics and receive good medicine and proper health care. Some people go to government hospital because they can't afford private hospitals. In government health facilities doctors do not attend the patient properly and their attitude is not good. Medicines are unavailable"–(IDI-5)*

Some rural areas do not have any health facilities and residents are forced to travel long distances to urban localities. There was a unanimous opinion among the community that quality of care needs to be significantly improved at public health facilities, with 24-hour operational services for medical emergencies.

**Access to health providers.** We received varied opinions regarding accessibility of health providers. Some locals stated that access has improved, especially following awareness spread by LHWs on the importance of timely diagnosis and treatment. Conversely, others stated that access is hampered by the shortage of physicians, lack of hiring, absenteeism, limited female doctors and specialists. Since most physicians are not based in the area and must travel to rural TMK for work, they are not available for emergencies that occur after working hours which is a common scenario in rural communities.

**Barriers and facilitators.** Transport is a major barrier in accessing health facilities and providers. Emergency transport such as ambulances are not readily available, resulting in delayed treatment and increased risk of morbidity and mortality. Finances are also an important issue as local communities usually do not have reserve funds for emergencies. Community members stressed the importance of the need for focused efforts by relevant authorities to develop an efficient transport network to provide relief during a medical crisis.

*"Lack of transport is a major issue for us. During an emergency, like when a woman is in labor, it takes time to arrange transport. Additionally, car drivers also raise their rent in such situations"- (FGD-EM-M5)*

*"Oh, we face a lot of difficulties in reaching health facility. In emergency situations we hardly get a vehicle, sometimes we are out of money so we must borrow from someone else." (IDI-7)*

## Health systems

**Training and capacity building.** Physicians, nurses, CHWs and LHWs in the community shared the view that training and refresher courses are essential for professional development. They identified the need for better and more frequent training opportunities, particularly LHWs. Participants related that currently, doctors attend training workshops and only superficially brief LHWs who are the main source of health awareness and healthcare in the community. Additionally, present refresher courses are focused almost entirely on polio vaccination, and LHWs desired training on breastfeeding, complementary feeding, water, and hygiene as these are more frequent and important issues they are currently faced with in the community.

*"There is a lack of training sessions for LHWs. Mostly doctors attend trainings and then brief us (LHWs) in a superficial way"- (FDG-LHS-4)*

There is a significant shortage of vaccinators in rural TMK and LHWs are often included in vaccination campaigns to compensate for this shortage. The community gave mixed reviews regarding the frequency of visits and quality of services provided by LHWs. Most residents felt that LHWs are only concerned with the provision of vaccinations, especially the polio vaccine, and do not give due emphasis to hygiene, sanitation, water treatment and illnesses like diarrhea and pneumonia.

*"The LHWs do not provide us with any information about diarrhea and pneumonia. They are only concerned with the polio campaign."- (FGD-TMK-M7)*

However, some participants praised LHWs for their efforts in educating mothers about IYCF and the distribution of various supplements and medicines.

**Supply and logistics.** Medications are obtained from LHWs, PHCs, and hospitals. Along with frequent shortages of zinc supplements, ORS, and antibiotics, health facilities lack basic equipment and have frequent power outages. Cold storage for vaccines is available in very limited facilities and are frequently compromised due to power outages. There is an urgent need to improve supply chain efficiency by minimizing existing bottlenecks.

*"We advise people to go to government facilities, but people are going to private clinics because in government facilities medicines [antibiotics] are not available"–(FGD-BSK-5)*

**Health Management Information System (HMIS).** HMIS refers to a system of health data designed to generate information and facilitate evidence-based decision making. Participants identified it as an important tool that can be used to formulate health policy, improve supply chain management, and manage the spread of diseases.

*"HMIS can play an important role in health policy formation because we (LHWs) input a detailed report of births, deaths, diseases, remaining stocks and demand for supplies in the system"- (FGD-LHS-1)*

*"HMIS is helping us to control shortage of supplies"–(FGD-LHV-2)*

The community believed HMIS can be used to reduce the prevalence of diarrhea and pneumonia. Documenting cases can help policy makers understand the gravity of the problem in TMK and devise appropriate strategies for management. However, more training is required to ensure efficient utilization of the systems.

**Referral system.**  In TMK, referrals are common due to primary facilities lacking equipment and skilled personnel. Most residents avoid referrals due to financial constraints and transportation barriers. The present lack of transportation facilities incurs significant financial burden on residents, resulting in most refusing referrals despite the need for specialist care which is unavailable in majority of the villages.

*"Doctors were helpless about my child. My son required a ventilator but the hospital in TMK did not have it. We were referred to a hospital in Hyderabad, where my son got the required medical attention and became healthy"- (FGD-EM-M1)*

Some community members and health workers also discussed communication constraints and the attitude of doctors as a hindrance in the success of referrals.

*"Sometimes we [LHWs] refer patients with referral slips and doctors do not give proper attention to the patient. This practice causes a loss of trust within the community. If we had a communication system for referral and we could call the doctor to discuss, it will be good for us and our referred patients"–(FGD-LHS-2)*

*"Attitude of health facilitators at government hospitals are not friendly. There is no proper check-up due to increasing number of patients."–(IDI-7)*

According to the local community, this system can be improved by providing free transport and consultations for referred patients. Better communication between the patient and health facilities involved, via computerization of the referral system, is required. Most importantly, there is an urgent need to equip health facilities in TMK to deal with emergencies, which will reduce referral rates and enable people to receive treatment at lower costs from nearby facilities.

## Discussion

This study was conducted as part of the formative phase of the CoMIC trial, which provided insight into community preferences, allowing the design of trial interventions in accordance with the needs of the community.

The community residing in rural TMK belongs to a low socioeconomic background with limited access to basic amenities of life. The study highlights different perceptions that community members have regarding the importance of breastfeeding, timely and adequate complementary feeding, immunization and the use of ORS and other prescribed medicines. It is important to note that people belonging to low SES face many issues, most importantly earning a living, as most are daily wagers. Their focus towards survival leaves them with little or no energy to prioritize healthcare. Preventive measures are often ignored, and care is only sought in the event of disease or emergency. This explains why populations with low SES have comparatively poor health outcomes, especially women and children.

In this study, issues related to WASH, childhood illnesses, IYCF, and vaccination coverage were reported by the community. There was consistency between the data presented in our

study and findings from reported research. In TMK, water quality is not suitable for drinking or domestic use. Most people reported water in their area to be discolored, bitter, and emitting a foul odor. This corroborates with findings from a WHO report which suggested that approximately 70% of households are still drinking bacterially contaminated water [27]. Ahmed et al. reported TDS (33%), electrical conductivity (46%), chloride (34%), turbidity (27%) and hardness (11%) in the water samples collected from various districts of Sindh, exceeding the maximum acceptable values set by WHO and Pakistani National Environmental Quality Standards (Pak NEQS) [28]. Additionally, sanitation and hygiene conditions are dismal in the region. Most people do not have access to toilet and latrine facilities and therefore practice open defecation. This finding matches the data highlighting that two-thirds of the global population practicing open defecation lives in South Asia. The region has approximately 558 million open defectors living predominantly in rural areas [27]. Hand washing is also unsatisfactory in the community, as frequency is low and those who are compliant often wash without soaps. This finding does not match national level data which states that 69% of households used soap and water, and that only one in ten households did not have water, soap, or any other cleaning agent [11].

Issues related to WASH, coupled with low EBF and vaccine coverage increases the risk of contracting diarrhea and pneumonia. Pakistan has the third highest under-5 mortality burden for diarrhea and ARI [10]. Our results showed overall prevalence of diarrhea has reduced compared to previous years, but it remains a significant health issue in rural TMK. Most participants were aware of the causes, symptoms and prevention of diarrhea and pneumonia, and identified the importance of immunization. Similar care seeking behaviors for both diseases were seen, where the community preferred receiving appropriate medical treatment from doctors and LHWs. However, cultural practices and misconceptions were present, indicating the need for awareness activities and the proposed community mobilization and incentivization intervention to jointly address both diseases by improving health practices and care seeking behaviors.

We also found EBF practices were minimal. This corroborates evidence that in Pakistan only 4 out of every 10 babies were exclusively breastfed [29]. Vaccinations are also trailing behind the globally standardized targets. While acceptance of vaccination has increased, further efforts are required to reduce vaccine hesitancy and improve coverage. Vaccine preventable diseases present a substantial economic burden in Pakistan. For hospitalized cases of pneumonia, an episode can cost up to 235 dollars and a case of meningitis can cost over 2000 dollars [30].

Finally, our results showed that most people are interested in forming groups that use collective action to improve behaviors in society. Community mobilization programs and non-cash incentives such as construction of toilets and latrines, installation of hand pumps, and distribution of soaps were suggested to improve behaviors and health practices. This aligns with the evidence synthesis by Gilmore et al. which suggested that well-implemented and contextually-appropriate community engagement strategies can help prevent and control infectious disease outbreaks and epidemics [31]. A similar community-oriented initiative, The Social Mobilization for Empowerment (MORE) program, implemented in rural Pakistan in 2010 was designed to encourage self-help, collective action within the community and better linkages with the government [32]. The program was implemented as a large-scale randomized intervention in which residents in treatment villages were organized into grassroot organizations which appointed representatives to a village institution. These representatives had the authority to decide development priorities and allocate resources from village development funds [32]. This intervention was successful at community participation and showed that

social mobilization aimed at strengthening women's participation in collective action helps to improve public health outcomes [32].

Community engagement is a strategy that enables key stakeholders to take collective action towards a common goal. It has gained attraction as a strategy for addressing multifaceted problems [33]. Our study has several strengths and limitations. By conducting interviews with diverse stakeholders, we were able to collect in-depth data that provided a comprehensive picture of the communities' beliefs and attitudes regarding the nature, acceptability, and potential efficacy of an intervention. Our sample was diverse and therefore the patterns of care seeking behaviors can be generalizable to the entire TMK district. However, while we made significant efforts to minimize bias, the nature of the study may have allowed our results to be subjected to interviewer or recall bias. Additionally, data is restricted to TMK and may not be used to make population-level decisions.

Findings derived from this formative research were used to design the CoMIC trial, in which the intervention involved collective community-based conditional non-cash incentives. The eligibility to receive incentives was based on a composite indicator comprising of the following outcomes: age-appropriate fully immunized children, ORS use for diarrhea, and sanitation index. As advised by the results of this study, the incentives were based on need assessment and prioritization by the community themselves and were given only upon achieving improvement in the outcomes. The findings of this study were discussed with the local community and used to design the intervention in the CoMIC trial [34]. Through this incentivization and community mobilization trial, we aimed to improve behavior patterns and care seeking practices for childhood diarrhea and pneumonia in TMK.

## Supporting information

**S1 File. COREQ checklist.**
(PDF)

**S2 File. Inclusivity in global research checklist.**
(DOCX)

## Acknowledgments

We would like to humbly acknowledge the CoMIC team, whose unfettered devotion and hard work made this work possible. They deserve the real credit. We would also like to acknowledge the community of TMK for sharing their views with us.

## Author Contributions

**Conceptualization:** Jai K. Das, Zulfiqar A. Bhutta.

**Data curation:** Zahra Ali Padhani, Sultana Jabeen, Shaista Mughal, Shafaq Baloch, Manesh Gangwani, Karim Nathani.

**Formal analysis:** Faareha Siddiqui, Zahra Ali Padhani, Maryam Hameed Khan, Sultana Jabeen.

**Funding acquisition:** Jai K. Das.

**Investigation:** Mushtaq Mirani, Shaista Mughal, Shafaq Baloch, Imtiaz Sheikh, Sana Khatoon, Khan Muhammad.

**Project administration:** Mushtaq Mirani, Rehana A. Salam.

**Supervision:** Jai K. Das.

**Writing – original draft:** Faareha Siddiqui, Zahra Ali Padhani, Maryam Hameed Khan, Sultana Jabeen, Rehana A. Salam.

**Writing – review & editing:** Jai K. Das, Zulfiqar A. Bhutta.

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
