## [Decision Letter · Decision Letter 0]

29 Mar 2023

PONE-D-23-05271Health Behaviors and Care Seeking Practices for Childhood Diarrhea and Pneumonia in a Rural District of Pakistan: A Qualitative StudyPLOS ONE

Dear Dr. Das,

Thank you for submitting your manuscript to PLOS ONE. After careful consideration, we feel that it has merit but does not fully meet PLOS ONE’s publication criteria as it currently stands. Therefore, we invite you to submit a revised version of the manuscript that addresses the points raised during the review process. Please submit your revised manuscript by May 13 2023 11:59PM. If you will need more time than this to complete your revisions, please reply to this message or contact the journal office at plosone@plos.org. Please include the following items when submitting your revised manuscript:A rebuttal letter that responds to each point raised by the academic editor and reviewer(s). You should upload this letter as a separate file labeled 'Response to Reviewers'.A marked-up copy of your manuscript that highlights changes made to the original version. You should upload this as a separate file labeled 'Revised Manuscript with Track Changes'.An unmarked version of your revised paper without tracked changes. You should upload this as a separate file labeled 'Manuscript'.

We look forward to receiving your revised manuscript.

Kind regards,

Farooq Ahmed, PhD

Academic Editor

PLOS ONE

Journal Requirements:

   "The authors declare that the research was conducted in the absence of any commercial or financial relationships that could be construed as a potential conflict of interest."

Reviewers' comments:

Reviewer's Responses to Questions

**Comments to the Author**

1. Is the manuscript technically sound, and do the data support the conclusions?

Reviewer #1: Yes

Reviewer #2: Partly

2. Has the statistical analysis been performed appropriately and rigorously? 

Reviewer #1: Yes

Reviewer #2: N/A

3. Have the authors made all data underlying the findings in their manuscript fully available?

Reviewer #1: Yes

Reviewer #2: No

4. Is the manuscript presented in an intelligible fashion and written in standard English?

Reviewer #1: Yes

Reviewer #2: Yes

5. Review Comments to the Author

Reviewer #1: General comment: Manuscript is well managed and covered very important areas.

Introduction: Good

Methodology:Well explained

Results:Result chapter is elaborated well and is very informative

Discussion:very well written and author has well supported study results with the help of empirical literature

Reviewer #2: Thank you authors for this piece of work shading light on community mobilization and perceptions aimed to provide formative evidence for interventions aimed to reduce burden of diarhoea and pnemonia in Pakistan.

General or specific comments:

1. Please avoid using words or statements that blame population or participants throughout the manuscript. E.g. in the absyract, you said, " population fails to recognize...", please rewrite. "

2. Plus, please dont use non-research terms. E.g. ..."ill equipped..."....please write it!

Methods:

3. Separate data collection from analysis. Provide separate and detail report for both methods. The rigor of qualitative studies should be evident from detail reporting.

4. Sampling of participants is less reported. Please make it detail.

5. You mentioned two different method of data analysis: thematic analysis and content anlysis. Which one did you use. Please stick to one. Content anlysis is a design, not data analysis method. Write in detail how you did thematic anlysis. Was it open coding or priori coding approach you used during thematic analysis? How many levels of themes have been emerged from the data? Did you have subthemes or further categories of subthemes? It appears to be one level of theme reported in this study, please add more levels. This will help you to provide thick description for the themes and ensures understanding for the readers.

6. What do you mean by using two opposing approaches of research design: deductive and inductive by this same qualitative study? Qualitative appears to be inductive in general. So, please elaborate how this study implemented inductive approach. Please remove the idea of deductive. If you mean that you had some eliciting concepts for developinh stidy guides, please say so instead of mentioning deductive. It is not uncommon to have elicting concepts in inductive or qualitative studies.

7. Dont use " questionnaire" in qualitative studies. Please say study guides or tools.

8. Have a separate subsection for " Rigor" in the methods section. You need to elaborate the crediblity, depedendablity, transferrablity, and confirmablity of the study in that section. The ideas of prolonged engagement, tick deacription, refelexivity, peer debriefying, member checking, subjective neutrality,......many more techniques of qulity assurance throughout the study processes....need to be elaborated. Readers often use these points to judge the quality of your work.

9. Please rewrite the manuscript by using COREQ (consolidated reporing systle for qualitative study) guideline. You can find the gudeline in the PLOSEONE or any other journals.

Results

10. Table 2. Please avoid reporting individuals in the FGD. Use ranges.

11. Please provide subthemes and categories to the presented themes.

12. Please add more quotations as you provide subthemes to the themes

13. What is common and distinct for diarrhoea and pneomonia in this finding?

Discussion

14. Connected comment 13, how do the distiction inform the interventions differently?

15. Provide limitation of the study at the end of discussion

6. PLOS authors have the option to publish the peer review history of their article (what does this mean?). If published, this will include your full peer review and any attached files.

Reviewer #1: **Yes: **Najma Iqbal Malik

Reviewer #2: **Yes: **Yohannes Kebede, PhD

---

## [Author Response · Author response to Decision Letter 0]

1 Apr 2023

Response to Reviewer #2 comments:

We would like to truly thank the reviewer for going through the manuscript in such detail, and for all the relevant suggestions which have helped make the methods clear and manuscript strong.

1. Thank you for bringing this to light, we have rewritten all such lines.

2. Thank you, we have edited these terms.

3. We have now separated the two sections and added relevant details in both.

4. Edited to add more detail (Page 5-6).

5. Thank you for highlighting these important points.

We have now edited this and specified that we used thematic analysis.

We have included two levels of themes:

Themes: 1) Community characteristics and social networks, 2) community mobilization, 3) incentives, 4) childhood diseases, 5) IYCF, 6) immunization, 7) WASH, 8) access to health facilities and providers, and 9) health systems

Subthemes: Diarrhea and pneumonia (for theme#4), access to health facilities, access to health providers, barriers and facilitators (for theme #8), training and capacity building, supply and logistics, HMIS, referral system (for theme#9)

Some topics have subthemes while some do not. We have not added sub-themes where there was general information and creating sub-themes would dilute the core messages of the theme.

6. Yes, the reviewer is very right that we wrote deductive as some themes had been identified as they were present in the study tool (e.g., incentives, community mobilization, childhood diseases). Further themes and subthemes were created as data was coded and analyzed.

We have corrected and detailed the thematic approach in the “data analysis” subsection (Page 7).

7. Edited, thank you

8. Thank you for your detailed feedback. We have added a “Rigor” subsection (Page 8).

Extended fieldwork/ prolonged engagement

Triangulation of sources

Reliability

Peer review

Low-inference descriptors

9. Thank you, we have done this, and attached the COREQ checklist as Supporting Information, detailing where the required information can be found within the manuscript (non-track changed version).

10. In table 1 and line 168 (track changes manuscript), we have clarified the numbers included are for total FGDs and IDIs conducted, not individuals.

11. The figure shows the broad themes, while the detailed themes and sub-themes are mentioned in the draft. 

12. We have now added more quotations.

13. Added (Page 14)

14. Added (Page 23-24; line 571 – 576)

15. Added ‘Limitations’ towards the end of the discussion (Page 25).

---

## [Editor Report · Decision Letter 1]

4 May 2023

Health behaviors and care seeking practices for childhood diarrhea and pneumonia in a rural district of Pakistan: A qualitative study

PONE-D-23-05271R1

Dear Dr. Jai K Das,

We’re pleased to inform you that your manuscript has been judged scientifically suitable for publication and will be formally accepted for publication once it meets all outstanding technical requirements.

Kind regards,

Farooq Ahmed, PhD

Academic Editor

PLOS ONE

---

## [Editor Report · Acceptance letter]

8 May 2023

PONE-D-23-05271R1 

Health behaviors and care seeking practices for childhood diarrhea and pneumonia in a rural district of Pakistan: A qualitative study 

Dear Dr. Das:

I'm pleased to inform you that your manuscript has been deemed suitable for publication in PLOS ONE. Congratulations! Your manuscript is now with our production department. 

Kind regards, 

on behalf of

Dr. Farooq Ahmed 

Academic Editor

PLOS ONE